# Oral Assessment and Preventive Actions within the Swedish Quality Register Senior Alert: Impact on Frail Older Adults’ Oral Health in a Longitudinal Perspective

**DOI:** 10.3390/ijerph182413075

**Published:** 2021-12-11

**Authors:** Lisa Bellander, Pia Andersson, Helle Wijk, Catharina Hägglin

**Affiliations:** 1Department of Behavioral and Community Dentistry, Institute of Odontology, Sahlgrenska Academy, University of Gothenburg, 405 30 Gothenburg, Sweden; catharina.hagglin@gu.se; 2Centre for Gerodontology, Public Dental Service, Region Västra Götaland, 402 33 Gothenburg, Sweden; 3Department of Oral Health, Faculty of Health Sciences, Kristianstad University, 291 88 Kristianstad, Sweden; pia.andersson@hkr.se; 4Institute of Health and Care Sciences, Sahlgrenska Academy, University of Gothenburg, 405 30 Gothenburg, Sweden; helle.wijk@gu.se; 5Department of Quality Strategies, Sahlgrenska University Hospital, Region Västra Götaland, 413 45 Gothenburg, Sweden; 6Department of Architecture and Civil Engineering, Chalmers University of Technology, 412 96 Gothenburg, Sweden

**Keywords:** geriatric nursing, nursing home, older adults, oral care, oral health, preventive actions, quality register, register study, ROAG

## Abstract

Poor oral health is common among older people in nursing homes. To identify and prevent oral health problems among the residents, ROAG-J (Revised Oral Assessment Guide–Jönköping), a risk-assessment instrument, is used by nursing staff routinely, and the outcome is registered in the web-based Swedish quality register Senior Alert. This study aims to investigate the preventive actions registered when oral health problems are identified and the effect of these actions longitudinally. ROAG-J data registered at nursing homes in Sweden during 2011–2016 were obtained from the Senior Alert database. Out of 52,740 residents (≥65 years), 41% had oral health problems, of whom 62% had preventive actions registered. The most common action was “Assistance with cleaning teeth”. Longitudinally, during the five-year observation period, a slight increase in oral health problems assessed with ROAG-J was found. Registered preventive actions, however, led to significant improvement in the subsequent assessment for the ROAG items lips, tongue, and dentures. Standardised risk assessments like ROAG-J provide an opportunity to detect problems early and establish preventive actions. The study, however, indicates a further need for structured education and a continuous follow-up in ROAG-J. Moreover, increased collaboration between nursing and dental care to improve oral health for older residents at nursing homes is needed.

## 1. Introduction

Life expectancy in the world is increasing [1]. This applies not least to Sweden, which has a rapidly increasing older population with significantly more remaining teeth and much better oral health than in the past [2,3]. With a larger older population, the number of frail people also increases, and consequently, more and more older people are becoming dependent on care from society [4]. Frailty is strongly associated with an increased risk of developing oral diseases [5,6]. Thus, poor oral health is commonly seen in the residents in nursing homes in Sweden; approximately 70% have caries disease [7]. In the Swedish quality registry for caries and periodontal disease, between the years 2011 and 2020, a pronounced increase in decayed and filled permanent tooth surfaces (DFS) was detected from the age of 80 years. In contrast, a clearly declining DFS trend was found for the ages of 35 to 65 years [8]. Worldwide, the prevalence of untreated caries is also reported to peak in old age (after the age of 70) [9]. This shows that among the oldest dental patients, there has been a deterioration in oral health, which is probably due to frailty. For example, many of the medications used by frail older people have low salivary secretion as a side effect [10]. Dry mouth may result in the accumulation of food debris and dental plaque followed by the fast progress of caries due to prolonged oral clearance [11,12].

The oral health of older adults is of great importance for several reasons. Oral health problems may have a strong impact on the ability to talk, chew, and taste food and on the older person’s social interactions [13]. Oral problems can lead to pain, malnutrition, and reduced quality of life [14,15]. Risk factors like poor oral hygiene can result in oral infections, such as periodontal disease, root caries, and oral candidiasis [16,17,18]. Associations between oral health and several general diseases have been found and can impact the quality of life of older people [13]. Poor oral health may exacerbate existing diseases such as diabetes mellitus or promote new ones, like aspiration pneumonia [19,20]. Aspiration pneumonia is due to aspiration of bacterial pathogens from the oral cavity or upper gastrointestinal tract, often associated with impaired swallowing, and has been shown to lead to increased mortality [21,22,23]. 

Daily removal of dental plaque and debris from teeth is an important key factor in good oral health [24]. Good oral health and a high number of natural teeth without dental caries correlate positively with health-related quality of life [25,26]. However, the number of natural teeth seems to correlate negatively with oral hygiene. For example, it has been shown that older residents in nursing homes with more than 10 teeth were found to have a twice as high risk of poor oral hygiene [27]. The reasons could include difficulty due to heavily restored teeth and complex prosthetic constructions, motivation problems, cognitive and bodily impairment, and reduced oral function [28,29]. The older adult’s oral health may become dependent on the nursing staff having the time and ability to help perform daily oral care. However, assistance with oral hygiene from nursing staff has been reported to be insufficient [30]. One reason could be that they find it an unpleasant task due to care-resistant behaviour from residents because of, for example, cognitive decline. Another reason could be inadequate competence in oral health and/or negative attitudes to oral health care among nursing staff [31,32,33]. This, in turn, can depend on shortcomings in leadership and management control of the nursing homes. Further complicating the situation is the fact that when the older adult becomes frail and functionally dependent, a previously regular dental contact is often lost [34]. 

To help reduce the aforementioned oral health risks among people aged 65 years and older, the Swedish quality register Senior Alert (SA) added an oral health assessment instrument, ROAG (Revised Oral Assessment Guide), in 2011 [35,36]. In SA, ROAG was slightly modified and renamed ROAG-Jönköping (ROAG-J), as the city of Jönköping is the national centre for the register [37]. This assessment is primarily for use by non-dental-health professionals, like nursing staff in nursing homes, to enable them to detect whether the older adult has oral health problems. The nursing staff have often received a shorter theoretical training, about 2 to 4 h, in ROAG-J from dental professionals before they start using the assessment. This education may vary between municipalities. The validity and reliability of ROAG has in studies shown to be good [35,38,39]. In addition to the risk assessment, planned preventive actions or referral to a dentist or physician are also included in ROAG-J. Information about a patient’s risks and action plan are entered into the computer-based SA register. The quality register SA is nationally distributed in 288 of Sweden’s 290 municipalities. It is used in all levels of care of older adults, for example, in nursing homes, home health care, hospital wards, and health centres. In 2016, the coverage rate of SA in nursing homes was 78%, but for ROAG-J, the percentage was 50% [40,41]. The registry data can be used as a base for a prevention plan for an older person at risk and as a goal for quality improvement measures in care organisations, where patients’ oral health status can be followed over time, both in each nursing home or across many national nursing settings, and for research [37].

In our previous register study, the oral health status assessed with ROAG-J in individuals living in nursing homes throughout Sweden was studied from a cross-sectional perspective [41]. That study showed that the most care-dependent residents had poorer oral health status than less care-dependent residents. Therefore, it is important to follow up and evaluate this effort in nursing care and its oral health effects. The aim of this study was to investigate the longitudinal effect on the oral health of older people when using ROAG-J in Senior Alert for risk screening in nursing homes from a national perspective. A further aim was to examine to what extent preventive actions are registered when oral health problems have been assessed by ROAG-J and what actions are taken. 

## 2. Materials and Methods

### 2.1. Design and Sample

The study is a descriptive, longitudinal, retrospective register-based study. The data were obtained from the national quality register SA. 

The study population consists of adults in Sweden aged ≥ 65 years who:lived in nursing homes, including special housing for people with neurocognitive disorders;had at least two ROAG-J assessments registered in SA with one-year interval, between the years 2011 and 2016.

### 2.2. Data Collection

Data in the present study were ordered from the Swedish Quality Register Center, Uppsala Clinical Research Center (UCR), in 2018. SA changed its database structure in 2017, and data from 2017 and later could therefore not be merged with 2011–2016 data for data analysis. The data file from SA included 105,102 individuals, of whom 52,362 individuals had only one ROAG-J assessment registered between the years 2011 and 2016 and were therefore excluded.

The ROAG-J assessments on the residents are performed by nursing staff. ROAG-J should be made at admission to the nursing homes and repeated at least twice a year. The examinations are both intra- and extraoral and take approximately 4 min to perform. There are nine items screened for in ROAG-J. The scoring is Grade 0–3 or 1–3, see Table 1.

The score for examinations and registered actions is entered into the database of the quality register SA. A grade 2 or 3 in one or more items is in the present study referred to as “oral health problems” or “risk”. SA recommends that an action programme should be planned if risk is detected in ROAG-J, and after a maximum of three months, a follow-up should be made [37]. A registered grade 2 means that a prevention action plan should be set up for the item or items in question, and these preventive actions are then to be carried out by nursing staff. When severe oral problems (grade 3) are registered in ROAG-J, a referral to or contact with a dentist or a physician is recommended [37]. Therefore, in this study, separate subgroup analyses are performed when a grade 3 in at least one item has been registered, presented under the heading “severe risk”. However, grade 3 is also included together with grade 2 when risk is presented in total or on item level. Figure 1 shows the preventive care approach of oral health in SA.

During the study period, a total of 69 different preventive oral actions were registered in the data file from SA. Many measures were very similar and were therefore merged by the authors, resulting in eight categories:Contact or referral to a dentist/physician when at least one grade 3 is registered;Assistance with cleaning teeth;Assistance with cleaning mucous membranes, tongue and dentures;Extra fluoride in addition to ordinary toothpaste;Pain relief for lips and/or oral cavity;Saliva substitute or moisturising/lubrication of mucous membranes and/or lips;Information, instruction and motivation regarding oral health and/or oral hygiene;Other oral care measures, such as facilitating practical measures influencing diet and/or swallowing.

Even if several preventive actions have been registered for one person within a category, in the statistical analysis, they count as only one preventive action per category. In table and text, individuals who had at least one preventive action registered in SA are presented as “Registered actions”, and individuals who declined actions or who had no preventive action registered in SA are presented as “No registered actions”. 

In addition to the assessment of oral health, the SA register also includes assessments for detecting and preventing falls, pressure ulcers, malnutrition, and bladder dysfunction. Variables from two of these assessments were included in the present study when reporting characteristics of the study population: “Physical condition” from the modified Norton and “Neuropsychological problems” from the Minimal Nutrition Assessment–Short Form [42,43].

### 2.3. Statistical Methods

Results are shown in numbers, percentages, means, range, standard deviation (SD) and *p*-values. For comparison of registered actions between groups when oral health problems (“risk”) were detected, Fisher’s exact test was used for gender, and Student’s t-test for age (continuous scale). For the comparison, the risk between the first and the subsequent ROAG-J assessments and registered actions/no registered actions, Fisher’s exact test was used. All tests were two-tailed (α = 0.01). All statistical analysis was performed using SPSS version 25 (IBM Corp., Armonk, NY, USA).

### 2.4. Ethical Consideration

The study was approved by the Regional Ethical Review Board of Gothenburg, Sweden (Dnr. 026-18). All older adults included in the quality register SA have been informed and have approved their registration there. Even though the register includes data on frail older persons, we judge the risk of harm as very low since the data for this study is retrospective and anonymous, with no personal data included in the data file. 

## 3. Results

### 3.1. Sample

The study population consists of 52,740 individuals from nursing homes in Sweden. During the period 2011–2016, they had all received a first ROAG-J assessment and at least one more the following year. The mean age was 85 years (SD 7.4; range 65–109), and the majority were women (68%). Characteristics of the study population are shown in Table 2.

Oral health problems (risk) were detected in 41% (n = 21,394) of the older people at the first assessment. Men had somewhat more oral health problems registered than women (43% vs. 39%). The youngest age group had the most oral health problems (65–74 years: 47%, 75–84 years: 42%, 85–94 years: 39% and ≥95 years: 38%). Severe oral health problems (at least one grade 3 risk) were detected in 12% (n = 6147) of the residents. “Severe risk” was more common for men than women (13% vs. 11%) and in the younger age groups (65–74 years: 17%, 75–84 years: 13%, 85–94 years: 10% and ≥95 years: 10%). “Risk” and “severe risk” detected in each item in ROAG-J are shown in Table 3. 

### 3.2. Planned Registered Actions

In individuals with identified risk at first ROAG-J assessment (n = 21,394), 62% (n = 13,285) had at least one planned preventive action registered. Among these, on average, 1.8 planned action categories were registered. The action “The person declines actions” was registered for 2% (n = 420) of the individuals with risk, and 36% (n = 7689) had no registered actions despite the assessed risk. For individuals with no registered oral health problems (n = 31,346), 13% had a planned preventive action registered, anyway. Figure 2 shows the percentages of planned actions per item and in total when oral health problems were found at the first ROAG-J assessment.

In the subgroup with “severe risk” (n = 6147), 63% (n = 3848) had at least one planned action registered, but the recommended action for grade 3, “Contact or referral to a dentist/physician”, was only registered for 13% (n = 772). For individuals with risk, a significant difference (*p* = 0.001) was found for age and registered preventive actions, the two older age groups having somewhat fewer actions registered compared to the younger age groups (65–74 years: 63%, 75–84 years: 64%, 85–94 years: 61% and ≥95 years: 60%). No difference was found for registered actions and gender (both men and women: 62%). Percentages and data analysis regarding registered planned actions by gender and age are shown in Table 4. Among the 21 regions in Sweden, the proportion of planned actions for those with risk varied from 41% to 77%.

### 3.3. Difference between the First and the Subsequent Assessments

Table 5 shows the oral health problems detected in ROAG-J for the first and subsequent assessments. There was a slight increase in the number of residents with detected oral health problems between the first and the next year assessment (41% vs. 43%, p < 0.001). This result is also reflected for the residents with two to four subsequent ROAG-J assessments (Table 5). 

Among residents with one subsequent assessment (n = 52,740) and a registered risk in the first assessment (n = 21,394), 22% became “better” and had no risk in the next year’s assessment. Among those with no registered risk (n = 31,346) in the first assessment, 18% became “worse” and had risk the second year (Table 6). Even though a higher percentage became “better” than “worse”, the higher number of individuals in the “worse” group resulted in the slight increase in individuals with ROAG-J risk each year seen in Table 5.

Among residents with “severe risk”, 35% (1 subsequent assessment) to 62% (4 subsequent assessments) became “better” (had no grade 3) in the last assessment. Among those who had no “severe risk” in the first assessment, from 6% to 11% became “worse” (had grade 3) in the first to fourth subsequent assessments (Table 6).

Table 7 shows the difference between the first and the second assessment, depending on whether actions were registered or not when risk was identified. In total, no difference was found regardless of whether actions were registered or not; in both cases, 22% had no registered oral health problems in ROAG-J in the subsequent assessment. Neither were there, in connection to risk, any differences found for gender and age when action had been registered or not, or when several subsequent assessments had been performed. On item level, for “lips”, “tongue”, and “dentures”, significant improvement was found if actions had been registered than if not (Table 7). 

If the action “Contact or referral to a dentist/physician” was planned in connection with the registration of “severe risk”, this resulted in statistically significant improvement, in contrast to no action being planned (Table 5). On item level, significant improvement was found for “teeth” (42% vs. 34%, *p* = 0.004) and “dentures” (42% vs. 20%, *p* < 0.001), if this action was planned.

## 4. Discussion

This study examines the longitudinal effect on oral health in just over 50,000 nursing home residents in Sweden who were examined with ROAG-J by nursing staff within the quality register SA. Senior Alert has been successfully implemented in many nursing home settings nationwide, and about half of all residents have been assessed with ROAG-J [40,41]. 

A main result of the present study was that preventive actions were not planned for about 40% of residents, despite detected oral health risk. The most common action was “Assistance with cleaning teeth”, where about half of the individuals with risk on items “teeth” or “gums” had the action registered. However, even though this is not an ideal result, it still indicates an improved awareness of the importance of oral health. A former study where 22,000 nursing home residents were examined (not using ROAG) by dental hygienists in Sweden showed that 77% had unacceptable oral hygiene, and only 7% of them were receiving assistance from nursing staff with daily oral hygiene [30]. Other national studies have shown that only 19% of older people aged ≥65 years at short-term care units received help from care staff with daily oral care, and only 9% of older adults in municipal residential care with risk of malnutrition registered in SA had oral care as a planned intervention [14,44]. In contrast to these studies, the result of the present study shows that nursing staff are quite prone to register that assistance with daily oral care is needed if the older residents have been assessed to have oral health problems. Although we cannot be sure it is put into practice, it suggests that the introduction of the oral health risk assessment in the quality register SA can improve the healthcare professionals’ awareness of the importance of supporting oral preventive care in dependent older adults. 

It has been reported that 70% of the residents in nursing homes had caries, especially root surface caries [7]. It is well known that increased fluoride administration is the most important part of preventive measures to avoid caries, especially for older people with salivary hypofunction [45,46]. For example, the use of higher-dose fluoride toothpaste (5000 ppm) has been reported to be significantly more effective than ordinary toothpaste (1450 ppm) in controlling caries progression in nursing home residents [47]. It is, therefore, noteworthy that only 6% of the individuals in the present study who were detected as having oral health problems received extra fluoride in addition to ordinary toothpaste. This result indicates a lack of knowledge among nursing staff about the importance of extra fluoride for care-dependent older people. It is important that the dental care personnel highlight the importance of fluoride administration in contact with and training of nursing staff and other health professionals. 

No improvement in registered oral health problems could be seen longitudinally between the years 2011 and 2016. One explanation may be that preventive oral health care actions were only registered for 13% of those without any detected risk in ROAG-J, which constituted a majority of the residents. Nursing home residents are a frail group with several risk factors that can lead to the rapid development of oral health problems [8,48]. This result suggests that even when frail older people are not assessed to be at risk for poor oral health, preventive actions need to be taken. In summary, oral health risk assessments that are recommended at least twice a year in SA are important for both those with and without risk. 

In total, there was no significant reduction in oral health problems registered in subsequent assessments, whether preventive actions were performed or not. Several reasons may explain this result. Lack of registered actions does not necessarily mean that the patient has not received any actions. Several care providers are also responsible per care recipient, not just the care provider who made the assessment and entered measures into the system. Thus, it is important to have good management and clear routines for the transmission of information between care providers in nursing homes. Another source of error is that it may be different individuals who perform the ROAG-J risk assessments the first time and at follow-ups, which entails uncertain reliability. However, ROAG has scored a good methodological quality of both intra-rater and inter-rater reliability [35,38,39]. However, the base of evidence is rather limited for all the existing oral health assessments for non-dental healthcare professionals, and policymakers need to be aware of these limitations when implementing them in healthcare and provide adequate education for their users [39]. 

Although no reduction in registered oral health problems could be seen in total, there were reductions on item level. Significant improvement was found for the ROAG-J items “lips”, “tongue”, and “dentures” at the second-year follow-up, if preventive actions had been performed, compared to not. The reason may be that preventive measures such as lubricating the lips and the tongue of another individual or brushing their dentures are easier to perform than brushing their teeth. The latter may be complicated due to complex prosthetic constructions and the fact that oral hygiene is considered a difficult task by nursing staff, who also often have limited time to assist residents [49]. There is also the ethical dilemma of residents not being able to comply with oral care due to impaired cognitive function. Other barriers to the provision of effective oral care have been reported, such as nursing home staff shortages, high staff turnover, the staff’s reluctance to comply with routines for residents’ oral hygiene, and their lack of knowledge of the importance of oral health for older persons’ general health and quality of life [27,50,51].

When severe (grade 3) oral health problems were found, followed by contact or referral to a dentist or physician, the following year’s registration showed a significant reduction in severe oral health problems. However, this action was only registered for 13% of older persons with registered severe oral health problems, despite being the recommended action in these cases. This result indicates that the collaboration between the care of older persons and dental care is important and needs to be improved. A report from 2019 by the National Board of Health and Welfare (NBHW) in Sweden concludes that dental care and health care lack a common holistic view of older patients, which leads to unnecessary and prolonged suffering for this group. The report emphasises that to achieve effective collaboration, systems for structured follow-up need to be developed. A starting point, suggested by NBHW, could be relevant national quality registries, like SA, where oral assessments can contribute to higher awareness of the association between oral health and general health in healthcare [52]. For oral care to be able to reach a central and more integrated position in nursing of older people, there is a need for improvements in many aspects, such as management’s involvement, standardised guidelines and quality assessments in the organisation, increasing care staff’s competence in oral care and close contact with and support from dental care [51]. Since the quality registry SA is a structured and already existing national system that is regularly used in the preventive oral health work in nursing homes, the data from the register can generate knowledge about the types of required improvements needed in nursing care. Thus, by including oral health in SA, a health-promoting initiative was taken that should improve the competence in and structure for oral health and oral care in the nursing of older adults and involve collaboration with dental care in a standardised way that did not exist before.

It is important to emphasise the communication between professional groups and their different responsibilities. Nurses in many institutions are involved in residents’ oral assessments, while daily oral care is provided by other care staff [53]. Sonde et al. (2011) reported that guidelines and routines for daily oral care in nursing homes are often non-existent and that the communication between nurses and other care staff could be increased with routine use of an oral assessment tool such as ROAG [49]. Basic knowledge of oral health and oral care for all health care professionals is an important goal to achieve good oral health [54]. The training of health care professionals to perform ROAG-J assessments by dental staff varies between nursing homes and regions in Sweden and is usually theoretical without practical elements. We, therefore, suggest that training in performing ROAG-J and education in providing oral health care for healthcare professionals should be developed and reworked in an organised way. Considering that behavioural problems, associated with, for example, cognitive impairment due to dementia and stroke, can significantly affect the outcome of good oral care, education of nursing staff is therefore suggested to extend beyond theoretical elements to also include both behavioural management strategies and hands-on practical elements [55]. 

In summary, there are many indications for the importance of examining frail older persons on a regular basis to identify oral health problems and to take preventive actions with the purpose to maintain general health and quality of life. In that work, ROAG-J in SA can be an important instrument, and non-dental staff have an important role.

## 5. Conclusions

Working with standardised oral health assessment, like ROAG-J in the quality register SA, provides the opportunity to detect risks and problems at an early stage and establish preventive oral actions in daily nursing care. This is crucial for maintaining good oral health among nursing home residents who are dependent on the actions of their caregivers. However, the study showed that preventive actions were not planned for about 40% of residents, despite detected oral health risk. This indicates that structures and routines in SA in terms of ROAG-J in nursing homes need to be improved, and the same applies to education and training. The results also show that the collaboration between nursing care and dental care is important and needs to be developed.

## Figures and Tables

**Figure 1 ijerph-18-13075-f001:**
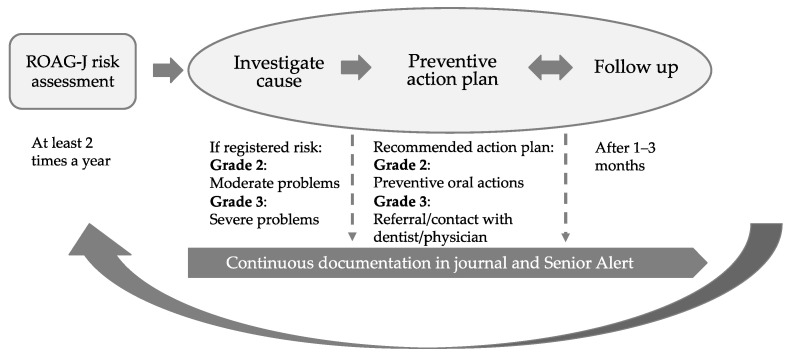
The preventive care approach for oral health (ROAG-J) in Senior Alert.

**Figure 2 ijerph-18-13075-f002:**
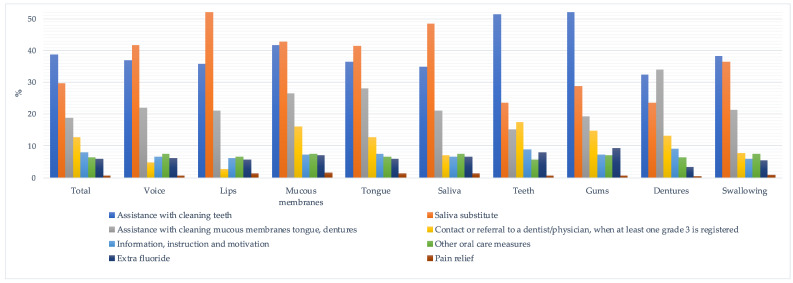
Percentages of registered actions if oral health problems (at least one grade 2 and/or grade 3) were detected at the first ROAG-J assessment, in total and for each ROAG-J item, except for the registered action “Contact or referral to a dentist/physician”, when “severe risk” (at least one grade 3) was detected.

**Table 1 ijerph-18-13075-t001:** ROAG-J in Senior Alert; the nine items and the grades ^1^.

Item	Grade 0	Grade 1	Grade 2	Grade 3
VOICE	Not applicable to judge	Normal	Dry, hoarse, smacking	Difficulty speaking
LIPS	-	Smooth, bright red, moist	Dry, cracked, sore corners of the mouth	Ulcerated, bleeding
MUCOUS MEMBRANES	-	Bright red, moist	Red, dry, or areas of discolouration, coating	Wounds with or without bleeding, blisters
TONGUE	-	Pink, moist with papillae	No papillae, red, dry, coating	Ulcers with or without bleeding, blistering
GUMS	No gums, only mucous membranes	Light red and solid	Swollen, reddened	Spontaneous bleeding
TEETH	No natural teeth	Clean, no visible coating or food debris	Coating or food debris locally	Coating or food debris generally, broken teeth
DENTURES	No prosthetics	Clean, functioning	Coating or food debris	Not used or malfunctioning
SALIVA	-	Runs freely	Runs sluggishly	Does not run at all
SWALLOWING	Not applicable to judge	Unimpeded swallowing	Minor swallowing problems	Pronounced swallowing problems

^1^ Grade: 0 = not relevant to assess, 1 = healthy or normal condition, 2 = moderate change or divergence, 3 = severe changes or divergences.

**Table 2 ijerph-18-13075-t002:** Characteristics of study population (n = 52,740).

	n	%
Age		
65–74	5485	10.4
75–84	17,610	33.4
85–94	26,020	49.3
≥95	3625	6.9
Gender		
Female	36,030	68.3
Male	16,710	31.7
Physical condition ^1^		
Good	26,504	53.3
Fair	20,939	42.1
Poor	2062	4.1
Very bad	234	0.5
Neuropsychological problems ^2^		
No problems	15,308	30.0
Mild dementia/depression	24,978	49.0
Severe dementia/depression	10,689	21.0

^1^. From the modified Norton, missing: n = 3001. ^2^. From the Minimal Nutrition Assessment–Short Form, missing: n = 1765.

**Table 3 ijerph-18-13075-t003:** Frequency and percentage of individuals assessed to have oral health problems (Risk or Severe risk) in each item in ROAG-J.

ROAG Item	Risk (Grade 2 and/or 3) n 21,394		Severe Risk (Grade 3) n 6147	
	n	%	n	%
Voice	4053	7.7	1230	2.3
Lips	3178	6.0	37	0.1
Mucous membranes	3269	6.2	257	0.5
Tongue	3372	6.4	86	0.2
Teeth ^1^	10,770	26.9	2568	6.4
Gums ^2^	4079	8.5	519	1.1
Dentures ^3^	3427	15.2	1376	6.1
Saliva	4133	7.8	172	0.3
Swallowing	5412	10.3	1237	2.3

^1^. Dentate individuals, grade 0 excluded. ^2^. Have gums, grade 0 excluded. ^3^. Denture wearers, grade 0 excluded.

**Table 4 ijerph-18-13075-t004:** Distribution (%), according to gender and age, of actions registered when risk (n = 21,394) was detected using ROAG-J. Fisher’s exact test (gender) and *t*-test (age) were used to analyse statistical differences shown with *p*-value.

	Gender			Age				
	Men n 7145	Women n 14,249		65–74 n 2575	75–84 n 7332	85–94 n 10,109	>95 n 1378	
Registered Actions When Detected Risk	%	%	*p*	%	%	%	%	*p* ^1^
Assistance with cleaning teeth ^2^	45.5	43.0	0.002	46.9	47.0	41.1	39.8	<0.001
Assistance with cleaning mucous membranes, tongue, dentures	18.8	19.0	0.725	16.3	18.4	19.4	22.4	<0.001
Extra fluoride	6.0	5.9	0.736	6.8	6.6	5.3	4.9	<0.001
Pain relief lips and/or oral cavity	0.3	0.4	0.186	0.4	0.8	0.7	0.6	0.884
Saliva substitute or moisturizing/lubrication	25.3	31.8	<0.001	27.7	28.8	30.3	31.8	<0.001
Information, instruction and motivation	8.6	7.6	0.018	8.9	7.5	8.1	7.7	0.382
Other oral care measures	6.2	6.6	0.274	6.1	6.4	6.6	6.0	0.693
Contact or referral to a dentist/physician ^3^	15.2	11.1	<0.001	13.4	13.2	11.5	13.5	0.103

^1^. Age as a continuous variable was used in the analysis. ^2^. Dentate individuals (grade 0 excluded from item “teeth”). ^3^. A grade 3 in at least one of the ROAG-J items.

**Table 5 ijerph-18-13075-t005:** Number of individuals in nursing homes who received two to five annual ROAG-J assessments (2011–2016) and change in ROAG-J risk, shown as number of individuals and percentage annually with risk and severe risk, respectively. *p*-values for risk–no risk between the first assessment and the last assessment with Fisher’s exact test.

Individuals n (%)	Assessments ^1^	Risk (Grade 2 and/or 3)			Severe Risk (Grade 3)		
		n	%	*p*	n	%	*p*
52,740 (100)	1	21,394	40.6		6147	11.7	
	2	22,440	42.5	<0.001	6733	12.8	<0.001
23,443 (44.5)	1	9380	40.0		2652	11.3	
	2	9639	41.1		2793	11.9	
	3	10,255	43.7	<0.001	3223	13.7	<0.001
7703 (14.6)	1	3126	40.6		880	11.4	
	2	3145	40.8		897	11.6	
	3	3218	41.8		994	12.9	
	4	3492	45.3	<0.001	1178	15.3	<0.001
999 (1.9)	1	364	36.4		100	10.0	
	2	376	37.6		108	10.8	
	3	403	40.3		135	13.5	
	4	416	41.6		137	13.7	
	5	429	42.9	<0.001	141	14.1	<0.001

^1^ As only 29 individuals had 6 annual ROAG-J assessments during the period 2011–2016, these assessments are not included in the table.

**Table 6 ijerph-18-13075-t006:** Number of individuals in nursing homes who received two to five annual ROAG-J assessments (2011–2016). Frequency and percentage of individuals who had risk in the first assessment and became “better” (no risk in ROAG-J in the last assessment), and those who had no risk in the first assessment and became “worse” (risk in ROAG-J in the last assessment), and those who were “unchanged” (still had risk or still had no risk in the last assessment).

		ROAG-J Risk (Grade 2 and/or 3)				ROAG-J Severe Risk (Grade 3)			
Individuals n	Assessments ^1^	Risk n		No Risk n		Risk n		No Risk n	
		Still Risk “Unchanged” n (%)	No Risk “Better” n (%)	Still No Risk “Unchanged” n (%)	Risk “Worse” n (%)	Still Risk “Unchanged” n (%)	No Risk “Better” n (%)	Still No Risk “Unchanged” n (%)	Risk “Worse” n (%)
52,740	1 2	21,394		31,346		6147		46,593	
		16,706 (78.1)	4688 (21.9)	25,612 (81.7)	5734 (18.3)	4009 (65.2)	2138 (34.8)	43,869 (94.2)	2724 (5.8)
23,443	1 3	9380		14,063		2652		20,791	
		6548 (69.8)	2832 (30.2)	10,356 (73.6)	3707 (26.4)	1398 (52.7)	1254 (47.3)	18,966 (91.2)	1825 (8.8)
7703	1 4	3126		4577		880		6823	
		2039 (65.2)	1087 (34.8)	3124 (68.3)	1453 (31.7)	405 (46.0)	475 (54.0)	6050 (88.7)	773 (11.3)
999	1 5	364		635		100		899	
		217 (59.6)	147 (40.4)	423 (66.6)	212 (33.4)	38 (38.0)	62 (62.0)	796 (88.5)	103 (11.5)

^1^ As only 29 individuals had 6 annual ROAG-J assessments during the period 2011–2016, these assessments are not included in the table.

**Table 7 ijerph-18-13075-t007:** Frequency of residents with registered ROAG-J risk in the first assessment and who either had actions registered or not. Percentage of individuals becoming “better” (no oral health problems) in the subsequent ROAG-J assessment. *p*-values are shown for analysis between groups (registered actions/no registered actions) with Fisher’s exact test.

ROAG-J Risk in the First Assessment		No ROAG-J Risk in the Subsequent Assessment				
	n	Registered Actions n	“Better” %	No Registered Actions n	“Better” %	*p*
Risk	21,394	13,285	22.0	8109	21.7	0.645
Voice	4053	2511	39.0	1542	36.0	0.057
Lips	3178	2068	49.0	1110	44.7	0.021
Mucous membranes	3269	2104	43.2	1165	42.3	0.658
Tongue	3372	2172	45.2	1200	40.8	0.014
Teeth	10,770	6909	26.9	3861	26.5	0.617
Gums	4079	2695	36.3	1384	36.4	0.918
Dentures	3427	2111	41.0	1316	36.8	0.015
Saliva	4133	2674	40.1	1459	38.2	0.244
Swallowing	5412	3269	28.1	2143	28.7	0.579
Severe risk ^1^	6147	3707	35.5	2440	33.7	0.147
Contact or referral ^2^	6147	772	41.7	5375	33.8	<0.001

^1^. Individuals assessed having “severe risk” (grade 3); any of the eight preventive actions can have been registered. ^2^. Individuals assessed having “severe risk” (grade 3) and the action “Contact or referral with dentist/physician, when a grade 3 is registered” having been used.

## Data Availability

Data in the present study were ordered from the Swedish Quality Register Center, Uppsala Clinical Research Center (UCR). The data can be obtained from the corresponding author on reasonable request.

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
