# Peer review of "Oral Assessment and Preventive Actions within the Swedish Quality Register Senior Alert: Impact on Frail Older Adults’ Oral Health in a Longitudinal Perspective"

_ijerph, 2021, doi:10.3390/ijerph182413075_

Round 1

Reviewer 1 Report

It is an interesting paper reporting on an oral assessment and preventive actions scheme in Sweden. Information shared by the authors may be useful for readers to develop or modify oral health schemes for older populations in the future. However, I have some comments which may help to improve the quality of the paper.

It is better to have a flowchart to show the screen, follow-up, and preventive registration procedures. 

The ROAG-J assessment was conducted by nursing staff. Any training sessions for the nursing staff before they started to use the assessment tool?

I was confused about the preventive oral action registration. Who was in charge of the registration categories? Based on what criteria? Please elaborate more on this issue.

The sample size of the present study is large. The authors adopted Fisher's exact test, which is commonly employed when the sample size is small. Although it is valid for all sample sizes, please justify the reasons for the employment of the statistic method.

The authors claimed that "Longitudinally during the five-year observation period, a slight deterioration in oral health was found (see Abstract)". The statement is problematic because oral health condition (e.g. caries, periodontal problems, and missing teeth) was not examined in this study and a lower grade of ROAG-J doesn't necessarily mean better oral health. Similarly, the presentation in Tables 4 and 5 is misleading. For example, "22% having better oral health in both cases (Page 9, line 296)", a lower score doesn't necessarily represent better oral health. Please revise the tone and words in the paper to avoid misinterpretation.

Reviewer 2 Report

This study aims to investigate the effect of the longitudinal effect of utilizing the ROAG-J on the oral health of older adults in nursing homes in Sweden and to examine the associated registered preventive actions. While the manuscript is interesting, many points should be addressed.

Introduction:

L39-40: none of the used references addresses the association between frailty and oral health. Add relevant references, please.

L42-51: While the fact that frailty could be a reason for changes in trends of oral health at old age is true. The authors missed a much important reason which is tooth retention that has increased as more older adults are retaining natural teeth into old age according to many epidemiological oral health surveys.

Check this paper:

McKenna, G., Tsakos, G., Burke, F. and Brocklehurst, P., 2020. Managing an ageing population: challenging oral epidemiology. Primary dental journal, 9(3), pp.14-17.

L63-65: provide references for these sentences.

Methods:

L182-183: “and those who had no preventive action registered or had 182 the action “The person declines actions” registered in SA, are presented as “No registered 183 actions”.”

The sentence is not clear Rephrase, please.

Results:

L201: were there any excluded participants? If yes, report, please.

L200: it would be useful to include a table that contains the sample characteristics and sociodemographics and perhaps health characteristics as well.

L220: “In individuals with identified risk at first ROAG-J assessment “

Provide the number of these individuals.

L224: “For individuals with no registered risk (n = 31,346) “

This sentence might confuse readers, I would use “no registered oral health problems” instead of “no registered risk”.

Table 2: I am not sure that the total number of individuals is accurate (n = 52,740) as individuals with no registered risk should have been excluded from the table. Check and correct, please.

L228: provide the total number of the subgroup “with severe risk”.

Table 3: provide the number of participants in the table heading.

 The analysis is confusing, as Table 3 shows that the comparison between registered actions and age were based on age groups ”categories”. So, I am not sure how the T-test was used to detect the differences for a categorical variable. Another point is related to the use of the Fisher exact test that is usually used with small sample sizes “which is not the case here”.

The layout of Table 4 is confusing and hard to follow. Moreover, the percentages in each up do not add up, the frequencies for participants with no registered risk and those with registered risk should be included.

Round 2

Reviewer 2 Report

I would like to thank the authors for incorporating my comments within the revised version of the manuscript. The manuscript has significantly improved. However, the authors still did not include relevant references that addresses the association between frailty and oral health. Check the following references:

Hakeem, F.F., Bernabé, E. and Sabbah, W., 2019. Association between oral health and frailty: A systematic review of longitudinal studies. Gerodontology36(3), pp.205-215.

Tôrres, L.H.D.N., Tellez, M., Hilgert, J.B., Hugo, F.N., de Sousa, M.D.L.R. and Ismail, A.I., 2015. Frailty, frailty components, and oral health: a systematic review. Journal of the American Geriatrics Society63(12), pp.2555-2562.

Author Response

The authors are very grateful for your suggestion. We have looked at the references that you so kindly suggested and have chosen to use these to address the association between frailty and oral health, see line 41-42.